# Positive Effect of Manipulated Virtual Kinematic Intervention in Individuals with Traumatic Stiff Shoulder: A Pilot Study

**DOI:** 10.3390/jcm11133919

**Published:** 2022-07-05

**Authors:** Isabella Schwartz, Ori Safran, Naama Karniel, Michal Abel, Adina Berko, Martin Seyres, Tamir Tsoar, Sigal Portnoy

**Affiliations:** 1Faculty of Medicine, Hebrew University of Jerusalem, Jerusalem 91905, Israel; isabellas@hadassah.org.il (I.S.); oris@hadassah.org.il (O.S.); 2Department of Physical Medicine & Rehabilitation, Hadassah Medical Center, Jerusalem 9765418, Israel; naamakar@gmail.com (N.K.); micsam@hadassah.org.il (M.A.); badina@hadassah.org.il (A.B.); martin.seyres@mail.huji.ac.il (M.S.); 3Orthopedic Department, Hadassah Medical Center, Jerusalem 9765418, Israel; tamirt@hadassah.org.il; 4Department of Occupational Therapy, Sackler Faculty of Medicine, Tel Aviv University, Tel Aviv 6997801, Israel; 5Physical Therapy Department, Hadassah Medical Center, Jerusalem 9765418, Israel

**Keywords:** virtual reality, biofeedback, shoulder pain, range of motion, motion capture

## Abstract

Virtual reality enables the manipulation of a patient’s perception, providing additional motivation to real-time biofeedback exercises. We aimed to test the effect of manipulated virtual kinematic intervention on measures of active and passive range of motion (ROM), pain, and disability level in individuals with traumatic stiff shoulder. In a double-blinded study, patients with stiff shoulder following proximal humerus fracture and non-operative treatment were randomly divided into a non-manipulated feedback group (NM-group; *n* = 6) and a manipulated feedback group (M-group; *n* = 7). The shoulder ROM, pain, and disabilities of the arm, shoulder and hand (DASH) scores were tested at baseline and after 6 sessions, during which the subjects performed shoulder flexion and abduction in front of a graphic visualization of the shoulder angle. The biofeedback provided to the NM-group was the actual shoulder angle while the feedback provided to the M-group was manipulated so that 10° were constantly subtracted from the actual angle detected by the motion capture system. The M-group showed greater improvement in the active flexion ROM (*p* = 0.046) and DASH scores (*p* = 0.022). While both groups improved following the real-time virtual feedback intervention, the manipulated intervention provided to the M-group was more beneficial in individuals with traumatic stiff shoulder and should be further tested in other populations with orthopedic injuries.

## 1. Introduction

Shoulder stiffness is defined as a restriction of the active and passive range of motion (ROM) of the glenohumeral joint [1]. A limitation in ROM means less than 100° range of motion in forward flexion [1]. Shoulder stiffness prevalence is estimated at 2–5% of the general population [2]. Other upper limb disorders might also limit ROM, e.g., damage following breast cancer treatment [3]. Shoulder stiffness can be secondary to a shoulder affliction, such as rotator cuff disease, osteoarthritis, trauma or surgery. In other cases, when the etiology of the stiffness is unknown, the condition is termed “primary frozen shoulder” or “primary idiopathic stiff shoulder” [1]. The loss of the shoulder’s full ROM may lead to significant impairment in functionality that reduces the ability of the patient to accomplish daily activities independently. Furthermore, altered kinematics occur as a compensation mechanism and may lead to the development of subacromial impingement, scapular dyskensia, tendinitis, and degenerative changes [4]. Interventions for this pathology often include physical therapy [5], anti-inflammatory medication, intra-articular hydrocortisone injections, distension arthrography, and surgery [6,7]. Home exercises may also be encouraged [8]. In a recent study that compared the clinical and cost effectiveness of three interventions for adults with frozen shoulder (early structured physiotherapy with a steroid injection, manipulation under anesthesia with a steroid injection followed by post-procedural physiotherapy, and arthroscopic capsular release followed by manipulation and post-procedural physiotherapy) [9], the authors reported that no conclusions regarding the superiority of one intervention over another could be drawn. While physiotherapy with a steroid injection is the most accessible option, manipulation under anesthesia is the most cost-effective option, compared to early structured physiotherapy or arthroscopic capsular release [9]. For all interventions, the main anticipated outcomes are increased ROM and reduced pain levels. 

As detailed in a recent review [10], individuals are more interested in leisure activities than in performing repetitive tasks during therapy. Consequently, virtual reality (VR) has been used and shown to induce repetition by enhancing motivation and enjoyment [10]. Since patients with shoulder stiffness might become uninterested in repetitive exercises and abandon them, thereby neglecting their rehabilitation, enriching exercises with newly available technologies may promote their motivation. For example, biofeedback systems have been shown to improve outcome measures in this population. In a recent randomized controlled study of 66 individuals with unilateral adhesive capsulitis, the study group performed shoulder abduction exercises with audible biofeedback provided by a wireless motion sensor [11]. The feedback volume was increased in relation to the elevation of the scapula. Compared to the control group, the study group that used the biofeedback showed increased scapular upward rotation and decreased shoulder pain and disability after two weeks and two months, respectively, and the improvement lasted six months [11]. Telerehabilitation has also evolved in recent years, mainly due to the requirement for social distancing following the outbreak of the COVID pandemic. A recent study showed that using inertial measurement unit-based sensors to track shoulder movements enhanced the three-month rehabilitation outcomes of individuals with stiff shoulder by increasing their compliance with training, thus improving functional recovery [12]. 

Virtual reality is another means to provide the user with real-time visual feedback of their performance. It has been used extensively for upper limb rehabilitation, e.g., using the Kinect sensor [13]. Different VR systems have been shown to be effective in survivors of stroke [14], as well as in individuals with kinesophobia and fragility in shoulder periarthritis [15]. To the best of our knowledge, it has yet to be applied for individuals with stiff shoulder. However, recently, a system that combines VR with wearable inertial measurement unit sensors was designed to perform motor assessment of shoulder ROM for this population [16]. The system was later utilized as a self-measurement system for shoulder joint mobility during four shoulder joint movements [17]. These systems have been used to provide non-manipulated real-time biofeedback. 

Importantly, VR allows influencing of a patient’s perception by manipulation of values presented to the user. This unique feature can provide a motivational addition to real-time biofeedback exercises [18,19]. For example, a study showed that when post-stroke individuals ambulated at their own pace on a treadmill connected to a VR visualization of a moving environment, when the VR optic speed was decreased, it promoted an increase in gait speed, although the participants were not consciously aware of it [18]. The authors assumed that if the patient walks faster during the physiotherapy sessions (even unintentionally), then the resulting high intensity exercise will shorten the rehabilitation period. However, the effect of manipulated virtual intervention for rehabilitation of the upper limb, specifically for individuals with traumatic stiff shoulder, is yet to be tested. Therefore, we sought to test the effect of manipulated virtual kinematic intervention on measures of active and passive ROM, pain, and disability level in individuals with traumatic stiff shoulder.

## 2. Materials and Methods

### 2.1. Study Design

This was a prospective randomized double-blinded study, in which individuals with stiff shoulder were randomly assigned to two groups, each receiving six sessions of exercises with kinematic biofeedback of shoulder movement. The feedback was manipulated for one group and not manipulated for the other. The group assignment was unknown to both the subjects and the researchers during the data collection. 

### 2.2. Population

We recruited 16 patients with stiff shoulder; however, three dropped out of the study of their own volition (see personal characteristics in Table 1). The inclusion criteria were: age 18 to 70 years, at 6–24 weeks following a fracture of the proximal humerus, with limited flexion and abduction up to 90°, and with normal or corrected eyesight. The exclusion criteria were: neurological pathology that affects the upper body, and previous shoulder injury or degenerative alterations to the shoulder that limit its ROM. Ethical approval was granted by the Hadassah Medical Center Helsinki Committee pretrial (approval number 0321–17-HMO, ClinicalTrials.gov Identifier: NCT03196674). All participants read and signed an informed consent form.

### 2.3. Measurement Tools

A goniometer was used to measure the active and passive ROM of the shoulder (flexion and abduction) before and after the intervention [20,21]. A visual analogue scale (VAS) was used to record shoulder pain [22,23]. In addition, the Disabilities of the Arm, Shoulder and Hand (DASH) questionnaire [24] was administered [25]. The DASH score ranges from ‘0’ (no disability) to ‘100’ (most severe disability). A subjective questionnaire concerning satisfaction with the intervention included nine questions rated from ‘1’ (not at all) to ‘5’ (very much so), including: I was pleased with the exercises, I was motivated by the exercises, the exercises contributed to my self-esteem, I felt comfortable during the exercises, the feedback I received during the exercises was clear, the exercise was easy, I succeeded in the exercises, I enjoyed the exercises, I would have liked to continue with these exercises during the rehabilitation. 

### 2.4. The Intervention 

The intervention took place at the Gait and Motion Laboratory at the Hadassah Medical Center in Jerusalem. Each subject received a 6-session treatment plan (2–3 times a week, 30 min per session). Each session included motion exercises of flexion and abduction. In each session, 11 reflective markers were placed on the following anatomic landmarks: right and left acromion, jugular notch, xiphisternal joint, T8 and C7 vertebras, medial and lateral epicondyles of the elbow of the injured limb, and a three-marker cluster placed on the injured humerus in the sagittal plane. Ten infra-red cameras (Qualisys, Gothenburg, Sweden) tracked the coordinates of the markers at a frequency of 120 Hz. The subject sat on a chair with no armrest, 2 m in front of a large screen (42 inch). Real-time visual feedback was provided using Visual 3D (C-motion, Germantown, MD, USA) showing a small skeletal representation of the subject’s torso and upper arm and a white moving graph showing advancing time on the *X*-axis and the 2D shoulder angle currently practiced on the *Y*-axis, i.e., either flexion or abduction. A yellow horizontal line was set at 90° to serve as a target (Figure 1). The kinematic feedback provided to the NM-group was the actual shoulder angle detected by the motion capture system. The kinematic feedback provided to the M-group was manipulated so that 10° were constantly subtracted from the actual shoulder angle detected by the motion capture system. The rationale for choosing a subtraction of 10° was based on preliminary trial-and-error with healthy individuals, where we surmised that, in the current settings of screen size and distance from the screen, subtraction of less than 5° might not be noticeable on screen, while a subtraction of more than 15° could be detected by the subject and he or she might be aware of a “mistake” in the feedback, so that the subject was not blinded to the manipulation. To achieve blinding of the researcher, four templates were prepared for combinations of flexion/abduction and manipulated/non-manipulated. The templates were named: ABD1, ABD2, Flex1, and Flex2. The subject received either an ABD1 and Flex1 combination or an ABD2 and Flex2 combination. Each combination was either manipulated or not, but this was not disclosed to the researcher collecting the data. 

### 2.5. Study Protocol

The subjects were randomly divided into two groups. One group received the non-manipulated feedback treatment (NM-group; *n* = 6) and the second group received the manipulated feedback treatment (M-group; *n* = 7). The shoulder passive and active ROM, pain and activity levels were tested at baseline and after the six sessions. The satisfaction questionnaire was filled out by each patient after the sessions.

### 2.6. Statistical Analysis

Statistical analyses were performed using SPSS 27.0 (SPSS Chicago, IL, USA). We performed a Shapiro–Wilk test and found that most of the parameters were not normally-distributed. We therefore represented descriptive statistics using median and interquartile percentages and chose the Mann–Whitney U test, a non-parametric test, to compare the two groups. The effect size, *r*, was calculated using the following equation [26]:(1)$$r=\frac{z}{\sqrt{N}}$$

Statistical significance was considered at *p* < 0.05. 

## 3. Results

There were no statistically significant between-group differences at baseline in the measures of active and passive ROM, VAS, or DASH scores (Table 2). However, following the intervention, the M-group showed greater improvement in the active flexion ROM and the DASH scores (Table 3). We used the Spearman correlation test to examine correlations between the pain levels and DASH scores before and after the intervention—no statistically significant correlations were found. 

As shown in Table 3 and Figure 2, measures of active flexion ROM and DASH scores were improved to a greater extent in the M-group compared to the NM-group. However, all the study population increased their active flexion ROM by at least 16°, which is the minimal clinical difference found for glenohumeral motion of people with a shoulder pathology [27]. However, only four (57.1%) subjects in the NM-group improved their DASH scores by more than 10.8 points, which is the minimal clinically important difference for this evaluation [28], while all of the subjects in the M-group improved their DASH scores by more than 10.8 points. 

There were no statistically significant between-group differences in the pain VAS scores. Only two subjects (28.6%) in the NM-group and one subject (16.7%) in the M-group improved their VAS score by more than 3 cm, which is the clinically important change in pain VAS score [29].

There was no statistically significant difference in the satisfaction questionnaire scores (*p* = 0.712, *r* = −0.111), as the median and interquartile percentages were 33.0 (25.8–38.0) and 37.0 (29.8–40.5) for the M-group and NM-group, respectively.

## 4. Discussion

In this double-blinded pilot study, we showed, for the first time, that a manipulated virtual real-time presentation of a subject’s movements during shoulder exercises promoted greater active flexion ROM and better DASH scores after six weeks of intervention. These results are consistent with the results of a similar manipulation of gait in stroke survivors [18]. To the best of our knowledge, this is the first study to show an effect of a manipulation-based intervention on functional measures of patients. Our pilot study may serve as a first step towards integrating a perceptual component into designed interventions that incorporate biofeedback to improve rehabilitation outcomes. 

The success of the manipulation of visual cues over the proprioceptive cues of the subjects in this study relies on recent findings from an investigation that incorporated a visuo-proprioceptive conflict [30]. In this investigation, reaching movements were viewed in a virtual environment and were altered in time (0.5 s delay). While the subjects were aware of the manipulation, the authors found that precision control determined the influence of separate sensory modalities on behavior, by biasing action towards cues from that modality. In our study, we anticipated that the subjects in the M-group would respond to the virtual visual feedback by increasing the ROM of either flexion or abduction. 

The greater improvement observed in active flexion ROM in the M-group is an encouraging finding in support of manipulated real-time kinematic feedback. A recent systematic review [31] concerning shoulder ROM needed for the performance of activities of daily living, showed that a shoulder flexion ROM of more than 90° is required for tasks such as turning a key, combing hair, or putting on a neckless. The majority of the subjects in both groups did not have adequate shoulder ROM to perform these activities. However, the improvement shown by our subjects, especially the significant improvement observed in the M-group, allowed them to extend their daily activities. Unfortunately, there were no between-group differences found in the active abduction ROM and the passive ROM. The baseline measures for the flexion ROM showed a trend for lower ROM of the M-group, although not statistically significant, so that some of the subjects in that group might have had a larger potential for improvement compared to some subjects in the NM-group. However, the highest value of active flexion ROM at baseline in the NM-group was only 92°, which was well below the normal active flexion ROM of 160° reported in a similar age group [32]. Therefore, we believe that the 54.6% median difference in percentage change in the active flexion ROM between the M-group and NM-group indicates an advantage of the manipulated feedback.

While some of the improvement in the active ROM might be attributed to compensation mechanisms via increased scapular upward rotation, clavicular motion, or position alterations [33], the overall results suggests that the M-group achieved improved forward flexion following the intervention, promoting their ability to perform daily activities, as indicated by the higher DASH scores of this group. Since the baseline DASH scores of the two groups were very similar, the statistically significant improvement in DASH scores of the M-group suggests a positive effect of the manipulated intervention. In addition to the improvement in DASH scores, shown by the higher percentage of change following the intervention, it is notable that all the M-group subjects improved these scores by more than the minimal clinically important difference, whereas in the NM-group three out of seven subjects did not improve their scores by more than the minimal clinically important difference. For these three subjects, the intervention was clearly less effective compared to its effect on the rest of the subjects. While there was no between-group difference in pain, we believe that the improvement in active flexion ROM resulted in improved functionality in daily activities, which is a primary outcome measure in patients’ rehabilitation. In cases where patients show no improvement in functionality following physical treatment, they become candidates for surgical intervention [34]. 

There was high between-subject variability in the percentage improvement in pain levels, as measured by the VAS scores. Although both groups showed similar improvement, only three subjects improved their VAS score by more than 3 cm. This might be explained by the use of conventional therapy in our medical center, as the pain level of each patient is respected. In a prospective multi-center study that compared the pain levels of individuals with stiff shoulder, it was found that patients undergoing conventional physiotherapy, where their pain threshold was respected, reported higher pain levels in the first few weeks of the treatment compared to patients who were encouraged to exceed their pain threshold [35]. Since our pilot study lasted only six weeks, the variability in the levels of pain improvement might be related to the effect of physiotherapy exercises conducted under painful conditions. A different method, in which subjects continue to pursue higher levels of ROM, despite their pain threshold being reached, might have produced different results.

Both groups reported similar satisfaction with the intervention. This is an important factor when considering the future integration of manipulated feedback in motor rehabilitation, since it shows that the blinded M-group was not aware of the perceptive alteration imposed by the feedback system. This indicates that the M-group performed non-volitional modulation towards higher intensity exercises, promoted by the reduced kinematics presented to them. This response proved to be advantageous for their rehabilitation. However, since one of the factors influencing patient satisfaction is clinical outcomes [36], the similarity of the satisfaction levels between the two groups seems to contradict the benefits of the M-group over the NM-group, as found in the active flexion ROM and DASH scores. This might be explained by other factors that can affect patient satisfaction, e.g., characteristics of the physiotherapist, patient features, the physiotherapist-patient relationship, and features of the healthcare setting [36].

The main study limitations are the small sample size and short intervention period. However, the effect sizes for both statistically significant findings, i.e., the active flexion ROM (0.555) and the DASH (0.634) score, exceeded a value of 0.5, which is considered a large effect size [37]. Since effect size is independent of the sample size and was found to be larger than 0.5 in this study, this supports the high impact of the intervention. Moreover, there was high between-subject variability in most measured parameters, which was expected for this population [38]. Future studies of manipulated feedback should involve a longer intervention and examine possible habituation effects of the perception manipulation. In addition, follow up examination to assess lasting effects of the intervention should be added to the protocol. 

## 5. Conclusions

We conclude that manipulated virtual kinematic intervention might be beneficial in individuals with traumatic stiff shoulder and should be further tested for other populations with orthopedic injuries, such as elbow and knee injuries. Furthermore, future system creators should construct VR systems for home-use (personal exercises and/or telerehabilitation exercises), in which kinematic manipulation is included in the exercises. Machine learning algorithms can be applied to negate habituation effects and increase the efficacy of treatment. The positive effect of biofeedback manipulation should also be investigated for other physical measures, such as grip or pinch forces and limb coordination. 

## Figures and Tables

**Figure 1 jcm-11-03919-f001:**
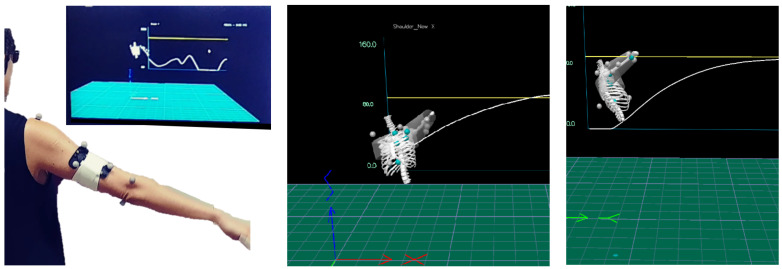
The subject (**left frame**) standing in front of the virtual presentation during shoulder abduction (**middle frame**) and flexion (**right frame**), as presented to the subject. The horizontal yellow line marks the target of 90°. The white line is the shoulder angle, presented in real-time. In these pictures, a healthy volunteer demonstrates a full range of motion.

**Figure 2 jcm-11-03919-f002:**
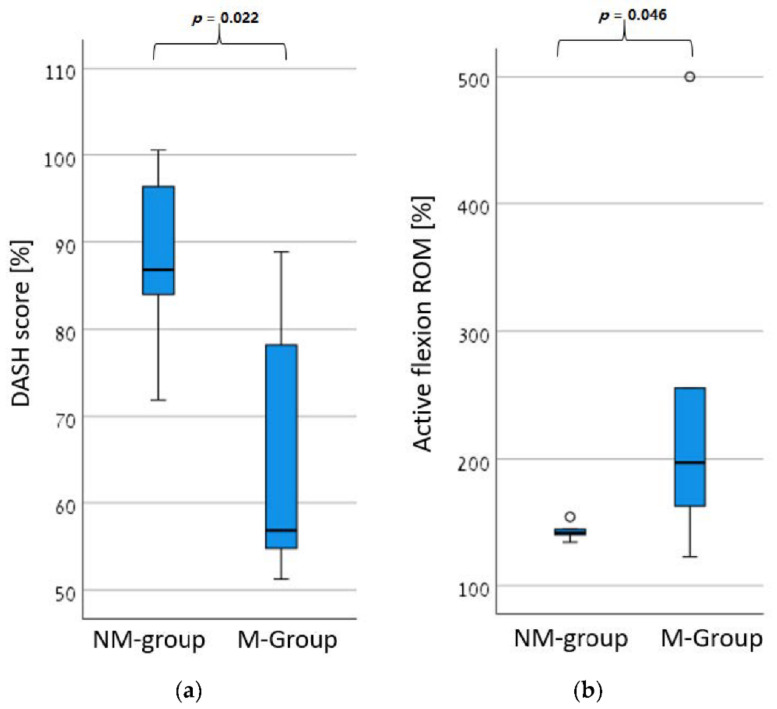
The percentage change (before and after the intervention) in (**a**) the Disabilities of the Arm, Shoulder and Hand (DASH) scores and (**b**) the active flexion range of motion (ROM) for both the non-manipulated feedback treatment group (NM-group; *n* = 6) and the manipulated feedback treatment group (M-group; *n* = 7).

**Table 1 jcm-11-03919-t001:** Patient characteristics of the non-manipulated feedback treatment group (NM-group) and the manipulated feedback treatment group (M-group).

Characteristic	M-Group (*n* = 6)	NM-Group (*n* = 7)	*p*
Age (years)	60.2 ± 6.1	63.3 ± 7.3	0.518
Sex	5 females, 1 male	7 females	0.261
Injured shoulder	2 left, 4 right	4 left, 3 right	0.391
Weeks from injury	6.2 ± 2.2	8.7 ± 5.1	0.563

**Table 2 jcm-11-03919-t002:** Baseline active and passive range of motion (ROM) of shoulder flexion and abduction, visual analogue scale (VAS) ratings for pain levels, and Disabilities of the Arm, Shoulder and Hand (DASH) scores. Values are presented as median and interquartile percentages for each of the two groups: the non-manipulated feedback treatment group (NM-group) and the manipulated feedback treatment group (M-group).

Characteristic	M-Group (*n* = 6)	NM-Group (*n* = 7)	*p*	*r*
Passive flexion ROM (°)	90.0 (70.0–100.0)	97.0 (92.0–105.0)	0.197	−0.358
Passive abduction ROM (°)	64.0 (47.5–75.0)	70.0 (60.0–80.0)	0.428	−0.220
Active flexion ROM (°)	61.5 (38.8–77.5)	82.0 (78.0–85.0)	0.053	−0.536
Active abduction ROM (°)	52.5 (39.5–59.0)	48.0 (44.0–60.0)	0.830	−0.060
VAS (0–10)	3.5 (0.8–5.1)	3.0 (0.0–5.0)	0.942	−0.020
DASH (0–100)	88.6 (72.7–108.6)	96.0 (81.1–104.3)	0.668	−0.119

**Table 3 jcm-11-03919-t003:** Percent of change (after the intervention/baseline × 100) in active and passive range of motion (ROM) of shoulder flexion and abduction, as well as the Disabilities of the Arm, Shoulder and Hand (DASH) score following 6 treatment sessions. The visual analogue scale (VAS) ratings for pain levels are depicted as difference (after the intervention - baseline). Values are presented as median and interquartile percentages for each of the two groups: the non-manipulated feedback treatment group (NM-group) and the manipulated feedback treatment group (M-group).

Characteristic	M-Group (*n* = 6)	NM-Group (*n* = 7)	*p*	r
Passive flexion ROM (%)	127.8 (113.3–173.2)	125.6 (106.0–152.5)	0.391	−0.238
Passive abduction ROM (%)	134.5 (115.8–191.4)	146.4 (137.3–180.0)	1.000	0
**Active flexion ROM (%)**	**197.1 (140.5–425.0)**	**142.5 (139.1–151.3)**	**0.046**	−**0.555**
Active abduction ROM (%)	150.0 (124.8–191.2)	162.5 (129.2–187.5)	1.000	0
VAS (0–10)	75.0 (12.5–106.8)	26.7 (5.0–120.8)	0.916	−0.034
**DASH (%)**	**67.7 (52.8–86.2)**	**89.7 (83.8–98.3)**	**0.022**	−**0.634**

## Data Availability

Data are submitted with the paper.

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
