# Peer review of "Positive Effect of Manipulated Virtual Kinematic Intervention in Individuals with Traumatic Stiff Shoulder: A Pilot Study"

_jcm, 2022, doi:10.3390/jcm11133919_

Round 1
Reviewer 1 Report
The authors present an interesting pilot study, you fool the patients with stiff shoulder, their movement dimensions are worse than in reality. The anteversion is significantly improved.
A total of 13 patients are tested in the M group.
The studies are cleanly structured and the presentation of the objectives, the implementation and the discussion of the results is comprehensible.
The reviewer has the question, why is a goniometer used to measure passive and active movement dimensions. The subjects all had 11 trackers during the exercises, which could have been used to determine the extent of movement much more accurately.
Author Response
Please see attached response.

Reviewer 2 Report
The application of VR to rehabilitation is of interest to everyone. This paper addresses this remarkable topic.
We believe that the findings regarding rehabilitation for periarthritis of the shoulder using VR are novel and of high academic value.
Below are my comments and questions.
1. Regarding Table 3, I felt it would be easier to understand if specific values for ROM, VAS, and DASH score were shown instead of rate of change. Also, please consider showing the change in values before and after the intervention in a graph.
2. We reduced the feedback for the M-group by 10 degrees from the actual angle, but please provide any rationale for setting the angle at 10 degrees.
3. Is the lower DASH score in the M-group due to pain?
4. The conclusions of this study regarding the usefulness of real-time feedback with VR (including methods to feedback reduced measurements) are appropriate. Since no statistically significant differences were found in the results of this study, we believe that caution should be exercised with regard to the description.
We have found some corrections to the description, which are shown below.
1. Page 1, line25 Does ‘manipulated intervention’ refer to both M- and NM-groups in terms of rehabilitation interventions? If it indicates the significance of real-time biofeedback exercises and manipulated feedback, I think the description needs to be revised.
2. Page 2, line 55 The word VR appears for the first time; there is an explanation of the abbreviation in line 72, but it should be explained here.
3. Page 2, line 83 ‘fed back’ seems to be a misnomer. Please check again for other typos.
4. In the abstract Page 1, line 24, ‘The M-group showed greater improvement in the active 24 flexion ROM (p=.046) and the DASH scores (p=.022). ‘ Is this appropriate word ‘greater’ when there is no significant difference?
Author Response
Please find attached response.
